# The Reflection Coefficient |r| as a Nondestructive Measure of the Coating Adhesion to a Steel Substrate

**DOI:** 10.3390/ma18194559

**Published:** 2025-09-30

**Authors:** Dariusz Ulbrich, Piotr Banas, Jakub Jezierski, Łukasz Warguła

**Affiliations:** 1Institute of Machines and Motor Vehicles, Faculty of Civil and Transport Engineering, Poznan University of Technology, 60-965 Poznan, Poland; piotr.banas@student.put.poznan.pl (P.B.); jakubjezierski00j@gmail.com (J.J.); 2Institute of Machine Design, Faculty of Mechanical Engineering, Poznan University of Technology, 60-965 Poznan, Poland; lukasz.wargula@put.poznan.pl

**Keywords:** reflection coefficient, ultrasound, coating, adhesion, car body

## Abstract

The main property of a steel substrate is the adhesion of its coating, which determines the quality and durability of the adhesive joint. The main objective of the research presented in this article is to evaluate the adhesion of coatings to substrates based on ultrasonic measurements and the determined reflection coefficient |r|. An experiment was carried out on disc samples, not only for ultrasonic measurements but also for the evaluation of the mechanical adhesion of coatings to substrates using the pull-off test. Three different methods of surface preparation of the samples were used: glass beading, surface treatment with P400 sandpaper, and the laser beam treatment. Based on the results, it was found that the best adhesion was obtained for samples with surfaces prepared by the glass-beading process. Reflection coefficient values in the range of 0.61–0.83 corresponded to mechanical adhesion in the range of 1.75–4.56 MPa. The results of the tests provide an important reference for the nondestructive evaluation of coating adhesion to substrates and allow for the estimation of mechanical adhesion based on the values of the reflection coefficient |r|.

## 1. Introduction

Adhesive joints, among other connections, such as welded [1,2], braze-welded [3,4], adhesive [5,6], spot-welded [7,8], and riveted [9] joints, are widely used in the construction and restoration of motor vehicles. Adhesive joints perform the important functions of both protecting the body sheet metal from corrosion [10] and providing a decorative effect [11], which is important to many users of road transportation equipment. Adhesive bonds, including the connection between a coating and its substrate, are characterized by properties such as adhesion, cohesion, and gloss [12]. Some solutions also provide resistance to UV radiation and are hydrophobic (provide drainage of water, resulting in no streaks and a better visual effect of the paint coating) [13,14]. Nevertheless, from the point of view of vehicle operation, one of the most important properties is the adhesion of the adhesive coating to the car body, whether it is a new coating (applied at the vehicle manufacturing plant) or a refinishing coating (applied during body sheet metal repairs).

The adhesion of a coating to a substrate is one of the many important properties of adhesive coatings that determine their service life and the quality of the joint [15]. Adhesion is the ability of a surface to adhere to another surface, that is, the surface bonding between two dissimilar bodies, resulting in a permanent connection [16].

Testing the adhesion of an adhesive coating to a substrate (usually steel) involves assessing how strongly the coating bonds to the substrate [17]. Research is often conducted by damaging the coating in a controlled manner and determining the load of the coating that peels off. In this way, the mechanical adhesion of the coating to the substrate is determined [18]. However, this type of testing is typically performed only as a last resort or on randomly selected vehicle components. This is because destructive methods damage the coating’s bond to the substrate, making the coated part unsuitable for further use or operation.

Therefore, nondestructive methods for the evaluation of the quality of coatings, including adhesion, are increasingly being sought to verify the condition of the connection without destroying it. Nondestructive testing methods used to assess the conditions of machine components and joints in the process of testing the adhesive joint include the ultrasonic [19], thermographic [20], eddy current [21], and X-ray [22,23] methods.

Ultrasonic quality assessment of adhesive joints is a nondestructive method that uses longitudinal wave propagation [24], surface waves [25], or guided waves [26]. It allows for the detection of defects, such as delamination [27], kissing bonds [28], and porosity [29]. All of these defects negatively affect the quality and durability of adhesive joints during their operation. Regardless of the waveform used or the ultrasonic testing technique applied, receiving a clear waveform from the adhesive joint area is crucial because it provides valuable information about the state of the adhesion between the coating and the substrate. Among the useful parameters of the ultrasonic wave for assessing condition, including detecting defects in adhesive joints, parameters such as the ultrasonic wave’s transit time, the average amplitude value in the frequency domain and the amplitude differences between individual pulses are used [30]. Another very important indicator, calculated based on selected properties of the ultrasonic wave, is the reflection coefficient |r| [31,32]. The value of this coefficient is inversely proportional to the adhesion strength—the lower the reflected energy from the joint boundary, the better the coating’s adhesion to the substrate. Ultrasound measurements allow for the assessment of adhesion distribution over the entire surface (detecting areas with weaker coating adhesion). The technique for testing and calculating the reflection coefficient is particularly valued for its nondestructive nature and its ability to accurately analyze the quality of the coating–substrate joint, which is important in quality control of coatings used, for example, in the repair of car bodies.

The main objective of the research presented in the article is to verify the adhesion of a coating to steel substrate based on ultrasonic testing and the determined reflection coefficient |r|, which was calculated by two different methods. The distinctiveness of the reflection coefficient |r| lies in its role as a nondestructive ultrasonic parameter that quantitatively characterizes the adhesion quality between the coating and substrate (steel, aluminum, and others). The smaller value of |r| corresponds to stronger mechanical adhesion of the coating to the steel substrate. This coefficient represents the proportion of the ultrasonic wave reflected at the coating–substrate interface, influenced by the presence and quality of the adhesive bond. In addition, to verify the ultrasonic results obtained and their correlation with mechanical adhesion, a pull-off test was carried out, allowing adhesion to be determined in MPa. The results of ultrasonic and mechanical adhesion tests for a particular type of coating (Novol 200 Hybryd car putty) using three surface preparation methods commonly applied body repair were compared. This is an important and novel addition to the existing knowledge in evaluating the adhesion of coatings to substrates for specific joint conditions. By evaluating the influence of distinct surface preparation techniques on the adhesion strength of this putty to the steel substrate, this research offers new insights crucial for the durability and quality of car body repairs. The use of ultrasonic testing as a nondestructive evaluation method allowed for the assessment of coating adhesion without damaging the coating or substrate, while mechanical adhesion tests served as a complementary validation method, confirming the reliability and precision of the ultrasonic measurements. The test results can be helpful for nondestructive evaluation of coating adhesion to a substrate and in quality assessment of post-accident car bodies after the repair process. Mechanical adhesion can be estimated based on the reflection coefficient results included in this article, taking into account the method of preparation of the steel substrate.

## 2. Materials and Methods

Coating adhesion tests using the reflection coefficient were carried out in several stages, according to the scheme shown in Figure 1.

The tests were carried out on disc specimens made of St3 steel, with a diameter of 50 mm and a height of 24 mm. The specimens had a notched M20 threaded pin in their central part, which enabled the pull-off test. This allowed the level of mechanical adhesion of the coating to the substrate to be determined. A view and diagram of the samples used in the tests are shown in Figure 2.

Surface preparation of the specimens was carried out by three different processing methods, including

Glass beading (abrasive blasting);Grinding with P400 sandpaper;Surface treatment at 60% of maximum power (1800 W).

These surface preparation methods were chosen based on the putty manufacturer’s recommendations and the authors’ own experience, reflecting the need to vary the adhesion of the putty coating to the steel substrate. For each method, 15 samples were used. In abrasive blasting, glass beads with a grain size of 0.4–1.0 mm were used. The surface treatment was performed using a gun equipped with a ceramic nozzle with a diameter of 5 mm, at an operating pressure of 5 bar. Another series of samples had their surfaces treated with P400 grit sandpaper mounted on an eccentric polisher. This surface preparation resulted in a smoother finish compared to the samples treated by glass beading. For the last group of samples, a laser beam of 1800 W was used, generated by an AccTek model AKQ-3000 laser (Jinan AccTek Machinery, Jinan, China). This method was selected because laser cleaning is increasingly used to clean surfaces, enabling rapid and efficient removal of corrosion, paint coatings, and other contaminants. The laser cleaning procedure, also known as laser ablation, removes oily deposits from the surface and eliminates oxidized layers and carbon buildup that can accumulate during the operation of machine and vehicle components. The specific parameters during the laser surface preparation of the samples were 1800 W power, a scanning frequency of 100 Hz, and a scanning width of 60 mm.

The surfaces of the samples prepared according to the technologies described above were verified by checking basic surface roughness profile parameters, such as R_a_ (the average surface roughness over the measurement length) and R_z_ (the average of the five largest total deviations recorded over the length). One measurement of these parameters was taken on the pin of each specimen at a measurement length of 5 mm.

For this purpose, a Taylor Hobson Surtronic 3+ profilographometer (Sutronic, Wegalaan, The Netherlands) was used. These measurements were necessary due to different surface preparation methods and allowed for verification of the similarity in the surface roughness profiles within each series of samples (using the same method of substrate surface preparation). This is extremely important from the perspective of mechanical adhesion and the anchoring of the coating within the surface micro-roughness, which directly affects the adhesion strength of the coating to the steel substrate.

The next stage of testing included ultrasonic amplitude measurements for the first three pulses from the bottom of the specimen (stage I) and from the coating–substrate interface area (stage II). Measuring the first three pulses was necessary due to the methods used for determining the reflection coefficient. Ultrasonic testing was carried out using the pulse-echo technique with a USM 35XS ultrasonic flare detector and a DS 12 HB 1-6 ultrasonic head at 6 MHz (both manufactured by Krautkramer, Cologne, Germany). The head was selected because its transducer width, and thus the ultrasonic wave beam, was 12 mm—slightly smaller than the diameter of the pin on which the putty coating was applied. Additionally, the head has a flat surface on which a 200 g weight was applied during the measurements to ensure constant pressure on the test sample. Three measurements were taken for each sample. This number was determined by preliminary tests, during which 20 measurements were taken of the uncoated sample, enabling calculation of the minimum number of measurements needed per sample. During ultrasonic testing in both stages, the amplitudes of the first three pulses were recorded (pulse 1—P_1_, pulse 2—P_2_, pulse 3—P_3_).

A diagram of the ultrasonic measurements for both the first and second stages of ultrasonic testing is shown in Figure 3. Additionally, it should be noted that ultrasonic testing was performed from the surface side of the pin that was not subjected to any surface treatment. The treated side was the opposite side from which the ultrasonic wave was reflected. Since the specimen surface was not modified in either stage I or II of ultrasonic testing (machining was performed earlier), this parameter was omitted from the evaluation of the ultrasonic results. Furthermore, it should be noted that the degree of surface roughness is a key factor in the adhesion of the coating to the substrate [33]. In ultrasonic measurements, surface roughness can also influence results depending on the type of ultrasonic wave used—for example, in the case of surface waves [25].

Between the stages of ultrasonic testing, a putty coating was applied to the previously prepared surface of the samples. This was the NOVOL 200 HYBRYD coating (Novol, Komorniki/Poznan, Poland), which can be applied to a variety of surfaces, including steel, aluminum, and galvanized steel. It is a modern multifunctional putty composed of a combination of glass microfibers and synthetic microsphere fillers, resulting in low volume shrinkage. The coating thickness was approximately 1 mm, consistent across all samples. The curing time of the coating was 25–30 min at 20 °C. During drying, a gradual color change was observed, from greenish when applied to a sandy color after curing. This change was due to chemical reactions between the putty base and the hardener (two-component putty). A view of the samples after the putty coating application is shown in Figure 4.

The visible unevenness of the putty coating on the sample surface was not significant. This relates to the coating thickness and its acoustic properties. The thickness (approximately 1 mm) was selected so that the ultrasonic pulse reflection occurred only at the joint boundary and not at the bottom of the coating. Part of the ultrasonic wave’s energy entering the car putty coating is absorbed into the material. The remaining energy returns to the ultrasonic head, providing information about the adhesion condition of the coating to the substrate. Additionally, as noted earlier, testing was performed from the pin side (the opposite side of the sample).

The next stage of the research involved calculating the reflection coefficient |r| by two methods, serving as nondestructive indicators of coating adhesion to the substrate [34,35]:Method I consisted of determining the reflection coefficient based on the difference between the amplitudes of the first echo measured in the first and second stages of ultrasonic testing, known as the total reflection method (Figure 5a);Method II consisted of determining the reflection coefficient using the three-echo method from the interface area (Figure 5b).

In both calculation approaches, regardless of the method applied, the algorithm considers the difference in amplitude of the selected ultrasonic wave pulses between the two stages of ultrasonic measurements. Figure 5 schematically illustrates the principle of ultrasonic measurements and recorded amplitudes of the selected ultrasonic longitudinal wave pulses.

The difference in amplitude between the first ultrasonic wave pulse measured during stages I and II of the ultrasonic testing was calculated using the following relation [34]:(1)$$∆W=20\times \log\frac{P_{1\prime }}{P_{1\prime \prime }}$$
where

ΔW—amplification of the ultrasound wave pulse;

P_1′_—the amplitude of the first (highest) pulse during the first stage of the measurement;

P_1″_—the amplitude of the first (highest) pulse during the measurement of the second stage, after the application of the coating.

This allows for the determination the reflection coefficient [35]:(2)$$\left|r\right|=10^{-\frac{∆W}{20}}$$
where

|r|—reflection coefficient;

ΔW—amplification of the ultrasound wave pulse.

The second method for determining the reflection coefficient considers the amplitudes of three echoes from the joint area. This is a modification of the bottom echo method, which does not account for the attenuation of the coating and substrate materials, nor the surface condition at the ultrasonic head contact point. The drawback of this method is the need to capture and isolate three distinct ultrasonic pulses. The reflection coefficient in this method is determined using the following relationship [34]:(3)$$\left|r\right|=\frac{1}{\sqrt{1+10^{\frac{{∆W}_{1}+{∆W}_{2}}{20}}}}$$
where

|r|—reflection coefficient;

ΔW_1_—amplitude drop between pulses P_1′_ and P_1″_, in dB;

ΔW_2_—amplitude drop between pulses P_1″_ and P_2″_ in dB.

The final stage of the test involves verification of the ultrasonic measurements by determining the destructive strength of the coating–substrate interface using a pull-off testing machine (Cometech B1/E, Taichung City, China). The pull-off test was followed by pin detachment, during which the maximum force required to cause adhesive damage at the coating–substrate interface was recorded. A jaw feed rate of 25 mm/min was applied during the testing. For all specimens, the destructive test duration did not exceed 30 s. Figure 6 illustrates the holder used in the pull-off test and its mounting in the testing machine jaws.

All of the above research stages enabled determination of the final test result, namely the reflection coefficient |r| and the correlation of these values with the mechanical adhesion of the coating to the substrate.

## 3. Results and Discussion

### 3.1. Roughness Profile Results

The roughness profiles results for two key parameters, R_a_ and R_z_, are presented in Table 1. All values fall within the typical ranges for the surface treatment methods examined, with R_a_ < 1.25 µm indicating a relatively smooth surface and R_a_ above 2 µm indicating a rougher surface. Examples of surface roughness profiles for selected samples, along with the measuring device, are shown in Figure 7.

For the glass beading results, the average R_a_ value was 2.41 µm with a standard deviation of approximately 0.42 µm, indicating a surface with relatively high roughness and inhomogeneity. The R_z_ parameter averaged around 12.35 µm with a standard deviation of 1.96 µm, reflecting the significant surface roughness, typical of this surface preparation method. Treatment with P400 sandpaper produced a much lower average R_a_ value of 0.72, with a standard deviation of 0.25, indicating a considerably smoother surface compared to the glass beading. The R_z_ value averaged 4.3 µm with a deviation of 1.29 µm, confirming the reduced roughness height, characteristic of abrasion with P400 sandpaper, which is commonly used for treating both the substrate surface and restoration materials during automotive body repair. The final method, laser beam surface preparation, resulted in an average R_a_ of 1.28 µm (standard deviation of 0.23 µm) and an R_z_ of 7.7 µm (deviation of 1.47 µm). These values suggest a surface smoother than that produced by glass beading but with greater irregularities than those treated with P400 sandpaper. It should be noted that the final choice of substrate preparation method depends on the requirements of the restoration materials manufacturers.

### 3.2. Ultrasonic Measurement Results and Reflection Coefficient Calculation

The detailed ultrasonic results, which served as the basis for calculating the reflection coefficient, are presented in Appendix A of this article. Additionally, Table 2 below summarizes the average amplitudes of the first three return echoes for stage I (before coating application) and stage II (after coating application). Standard deviations were also calculated to quantify the variability of the individual wave pulses amplitudes obtained during the ultrasonic tests. Although the third pulse from the specimen bottom and joint area was not used in the reflection coefficient calculations (depending on the chosen method), it was recorded to ensure accurate pulse identification in each ultrasonic measurement. This procedure helped verify the correct pulse sequence and obtain precise amplitude values for pulses 1 and 2, which are essential for accurate determination of the reflection coefficient using the proposed methods.

Based on the above results, it is important to note that the highest average amplitude values for the first pulse in stage I were recorded for the laser beam treatment, while the lowest values were obtained for surface treatment with P400 sandpaper. The same trend was observed in stage II of the ultrasonic test. The amplitude of pulse 1 in stage I for the glass-beaded surface was approximately 50% higher than that for the sandpaper-treated surface, likely due to differences in the substrate prior to putty application. Additionally, regardless of the surface preparation method, a decrease in the first pulse amplitude between stages I and II was observed, with the largest percentage decrease for the glass beading (~28.9%) and the smallest for P400 sandpaper (~23.3%). The laser beam treatment resulted in a decrease of approximately 25%, placing it between the other two methods. All three surface preparation methods showed a significant decrease in average amplitude between the first and second pulses, as well as the second and third pulses, regardless of the test stage. This indicates diminishing ultrasonic wave energy intensity as it propagated through the substrate. Standard deviations range between 2 and 9, reflecting relatively consistent measurement stability throughout the ultrasonic testing. Considering these results, longitudinal wave pulses amplitudes were calculated using two reflection coefficient methods. The average reflection coefficient values are presented in Table 3.

The average reflection coefficient values vary depending on the substrate surface preparation method. Using the total reflection method (Method I), the lowest mean value was obtained for samples treated by glass beading. Higher values of 0.75 and 0.79 were obtained for laser beam surface preparation and sandpaper grinding, respectively. For the method based on three-echo reflection, the coefficient values are lower but closely clustered, ranging from 0.52 to 0.55, indicating minimal variation regardless of the surface preparation technique. The differences in reflection coefficient values by the surface treatment method are approximately 36% for the laser beam treatment and sandpaper grinding, and 33% for glass beading. Standard deviations for all methods are relatively low, indicating good repeatability and low variability of the |r| coefficient.

In summary, all surface treatment methods show a notable decrease in the reflection coefficient depending on the calculation method. The second calculation method yielded results that are closer to each other, suggesting that the substrate’s material structure—which causes attenuation and dissipation of ultrasonic wave energy—significantly influenced the values determined by Method I.

### 3.3. Correlation Test Analysis

The final step in the study was to correlate the reflection coefficient results determined by the two different methods, with mechanical adhesion expressed in MPa and measured by the pull-off test. The results of these tests are shown in Figure 8, Figure 9 and Figure 10. Relating ultrasonic testing results to mechanical adhesion enables estimation of coating adhesion to substrates based on ultrasonic measurements and the calculated reflection coefficient. This correlation should consider the substrate preparation method, as it influences both the ultrasonic test results and mechanical adhesion.

For Method I of calculating the reflection coefficient |r|, the results ranged from about 0.6 to 0.9. The lowest values, between 0.61 and 0.83, were observed for samples with surfaces prepared by glass beading. This correlates with the highest mechanical adhesion values measured for these samples. Glass beading creates a rougher surface with more irregularities—peaks and valleys—that enhance coating adhesion. Samples prepared by the laser beam treatment showed reflection coefficient values between 0.72 and 0.81, corresponding to adhesion strengths ranging from about 1.47 to 3.5 MPa. These mechanical adhesion values were lower than those from the glass-beaded samples, which ranged from 1.75 to 4.56 MPa. For samples prepared with sandpaper, reflection coefficient values varied widely from 0.68 to 0.89, with mechanical adhesion between 1.82 and 3.39 MPa. The lower surface roughness of these samples means there were fewer small cavities for the coating to bond to, which reduced adhesion. Generally, greater surface roughness or cavity density increases adhesion, while the reflection coefficient values tend to decrease. Using the second method for calculating the nondestructive adhesion measure |r| yielded much lower values, mostly between 0.5 and 0.6. It is important to note that for the samples analyzed for the correlation between the reflection coefficient and mechanical adhesion, failure occurred adhesively at the coating–substrate interface. After the pull-off testing, no putty remained on the testing pin, confirming adhesive detachment (Figure 11). This analysis shows how the reflection coefficient |r| relates to surface roughness and mechanical adhesion strength among different surface preparation methods and aids in understanding interactions at the coating–substrate interface. In addition to lower reflection coefficient |r| values recorded for Method II, these results were less predictable, displaying no clear trend in ultrasonic parameter variation during measurement. This shows that when testing the coating adhesion to a substrate component of two completely different materials in terms of structure, it is important to take into account the acoustic properties of these materials, which affect the results of ultrasonic measurements. These parameters are taken into account in Method I, which measures the reflection coefficient based on the total reflection of the ultrasonic wave, first from the bottom of the sample (stage I) and then from the coating–substrate interface (stage II).

The proposed mathematical models relating the reflection coefficient and mechanical adhesion appear to be better matched for the total reflection method. This is supported by the correlation coefficient, which reached value of 0.8. In contrast, the correlation coefficient for the second method (the three-echo method) was significantly lower, likely due to the fact that Method II does not take into account the acoustic properties of the joined materials. Although the samples were made of a single material, the material structure may be heterogeneous, causing differences in the values of the tested ultrasonic parameters. For both reflection coefficient determination methods, a clear trend was observed where lower values of this parameter corresponded to higher coating adhesion to the substrate. This trend is confirmed by other research results available in the literature [36,37,38]. Furthermore, the obtained reflection coefficient results range from 0 to 1, confirming the accuracy of the performed calculations regardless of the method used.

The research results presented above provide a knowledge foundation for assessing the adhesion of adhesive coatings used in vehicle body repairs. By utilizing the reflection coefficient (and the relationships shown in Figure 8, Figure 9 and Figure 10), it is possible to estimate adhesion levels and identify areas with insufficient bonding, allowing for their removal during the repair process rather than later during vehicle operation.

## 4. Conclusions

Taking into account the results of ultrasonic and pull-off tests, the following conclusions can be drawn:The highest mechanical adhesion values, ranging from 1.75–4.56 MPa, along with corresponding reflection coefficient values |r| (0.61–0.83), were obtained for the glass-beaded samples surfaces. In contrast, the lowest values (1.82–3.39 MPa; 0.68–0.89) corresponded to samples whose surfaces were sanded with P400 sandpaper.For nondestructive assessment of car putty coating adhesion to steel substrate, Method I (total reflection method) is recommended due to its strong correlation with mechanical strength (R^2^ > 0.8). The optimal reflection coefficient |r| range for high-quality coating after glass beading is 0.61–0.83.Determination of the reflection coefficient using the three-echo method from the connection area is characterized by a much lower correlation coefficient in the range of 0.37–0.58 and lower reflection coefficient values (0.41–0.63).For the putty coating-steel substrate interface, the total reflection method for determining the reflection coefficient |r| combined with the mechanical adhesion results show a strong correlation—lower reflection coefficient values correspond to significantly higher mechanical adhesion of the coating to the substrate.The three-echoes reflection coefficient method shows a lower correlation with the mechanical adhesion results, primarily because it does not account for the acoustic properties of the materials forming the adhesive joint.

Based on these conclusions, the authors recommend continued research using nondestructive testing methods to evaluate coating properties, including adhesion to substrates. Therefore, the next step is to verify the proposed testing method using the reflection coefficient for other adhesive joints. Furthermore, efforts are planned to correlate fundamental roughness profile parameters with both reflection coefficient |r| and mechanical adhesion. In summary, these research results can provide significant support in the process of assessing the quality of coatings both during their production and during their use, which significantly enhances the application aspect of the presented research.

## Figures and Tables

**Figure 1 materials-18-04559-f001:**
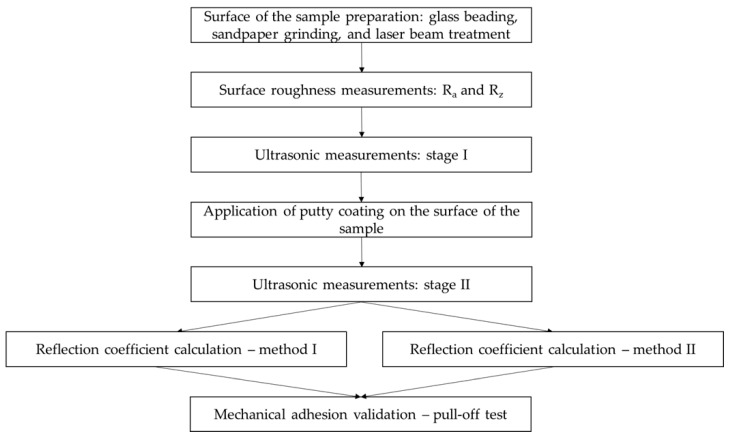
Stages of the experiment.

**Figure 2 materials-18-04559-f002:**
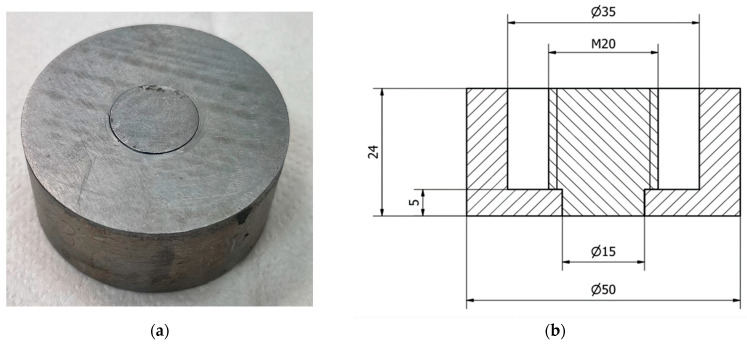
Samples used during experimental testing: (**a**) view of the sample; (**b**) diagram of the sample.

**Figure 3 materials-18-04559-f003:**
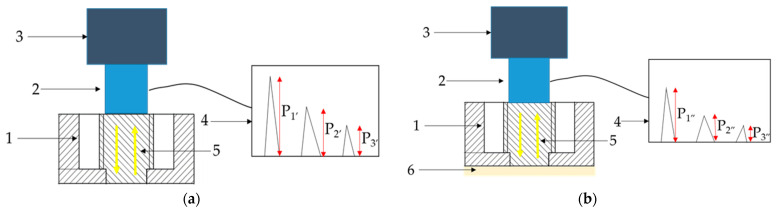
The concept of ultrasonic measurement; (**a**) before car putty coating application—stage I, (**b**) after car putty coating application—stage II; P_1′_, P_2′_, P_3′_—amplitudes of first, second, and third pulses in stage I of the experiment; P_1″_, P_2″_, P_3″_—amplitudes of first, second, and third pulses in stage II of the experiment; 1—sample, 2—ultrasonic head, 3—load, 4—flaw detector screen with pulses, 5—direction of ultrasonic wave propagation (yellow arrows), 6—car putty coating.

**Figure 4 materials-18-04559-f004:**
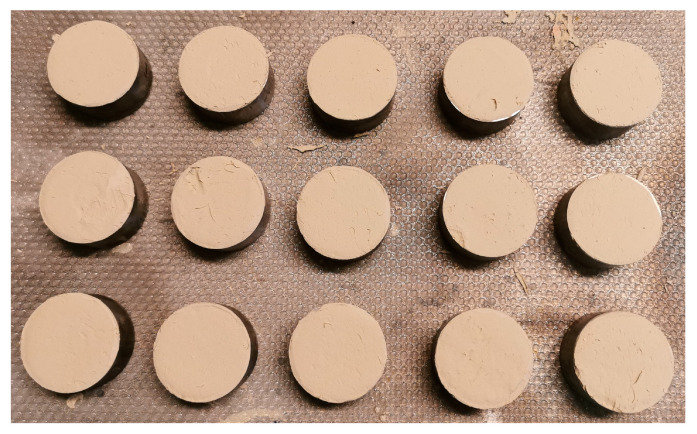
Samples after the car putty coating application.

**Figure 5 materials-18-04559-f005:**
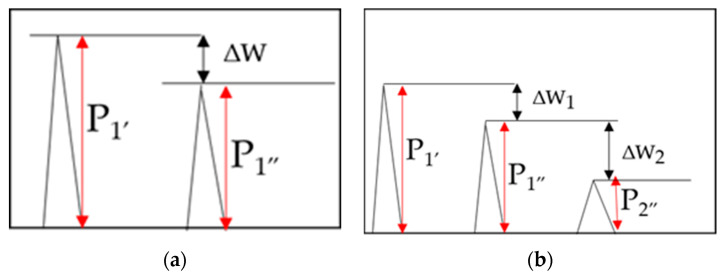
The concept of the reflection coefficient |r| measurement; (**a**) Method I; (**b**) Method II; P_1′_—amplitude of first pulse in stage I of the experiment; P_1″_, P_2″_—amplitudes of first and second pulses in stage II of the experiment, ΔW—amplification of the ultrasound wave pulse; ΔW_1_—amplitude drop of pulses P_1′_ and P_1″_; ΔW_2_—amplitude drop of pulses P_1″_ and P_2″_.

**Figure 6 materials-18-04559-f006:**
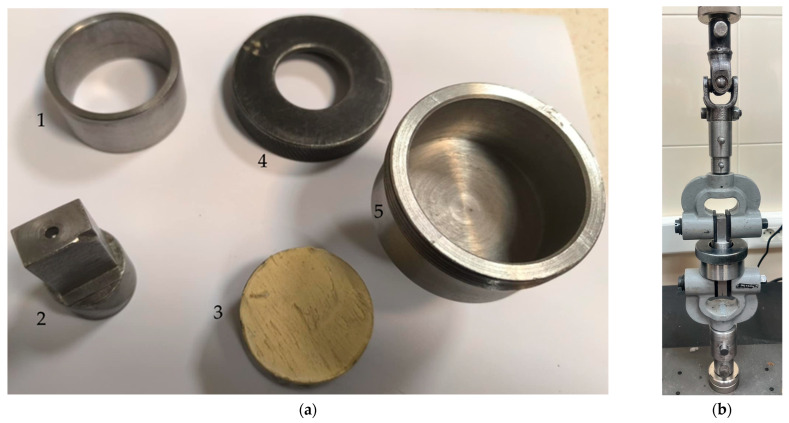
The concept of the pull-off test measurement: (**a**) components of the holder and sample; (**b**) holder with the sample during the pull-off test; 1—sleeve; 2, 4, 5—parts of the holder; 3—sample with car putty coating.

**Figure 7 materials-18-04559-f007:**
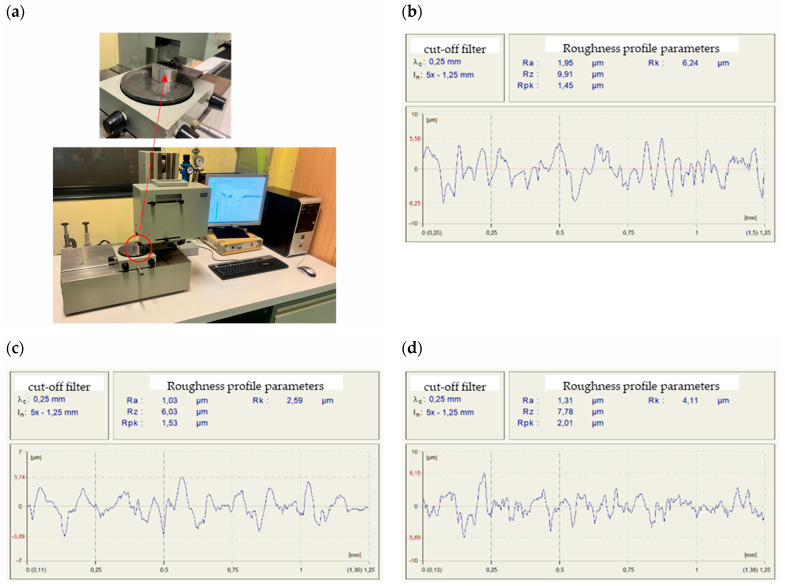
Roughness profile measurements: (**a**) measurement stand; (**b**) profile after glass bead treatment; (**c**) profile after sandpaper grinding; (**d**) profile after laser beam treatment.

**Figure 8 materials-18-04559-f008:**
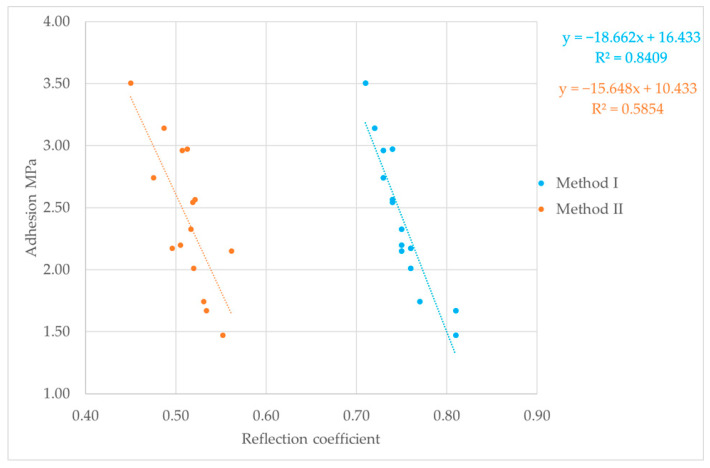
Reflection coefficient and pull-off test results for laser beam treatment.

**Figure 9 materials-18-04559-f009:**
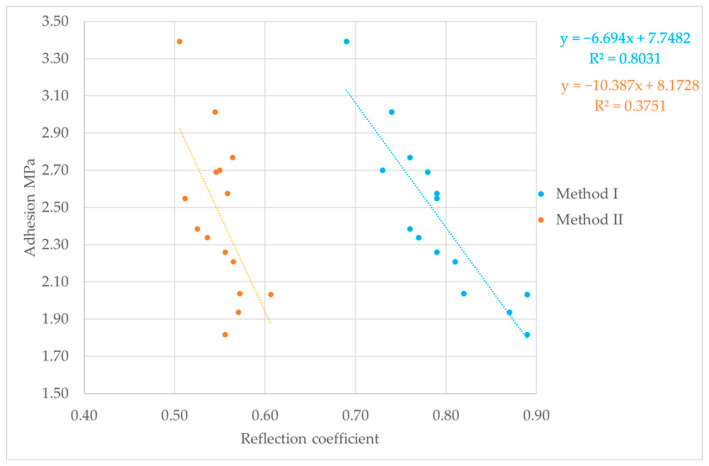
Reflection coefficient and pull-off test results for P400 sandpaper.

**Figure 10 materials-18-04559-f010:**
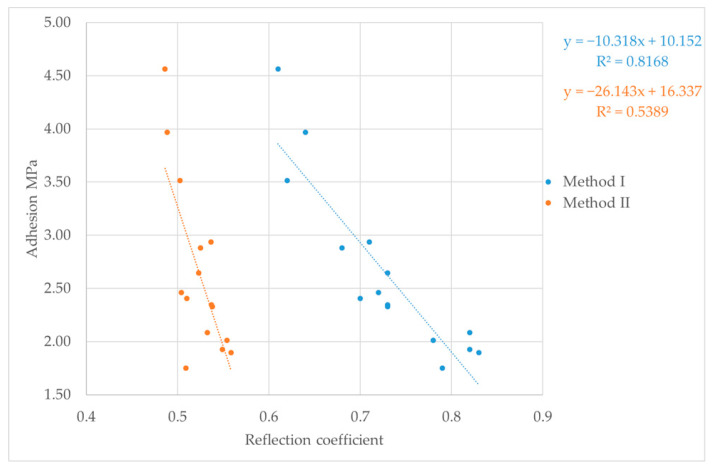
Reflection coefficient and pull-off test results for glass beading.

**Figure 11 materials-18-04559-f011:**
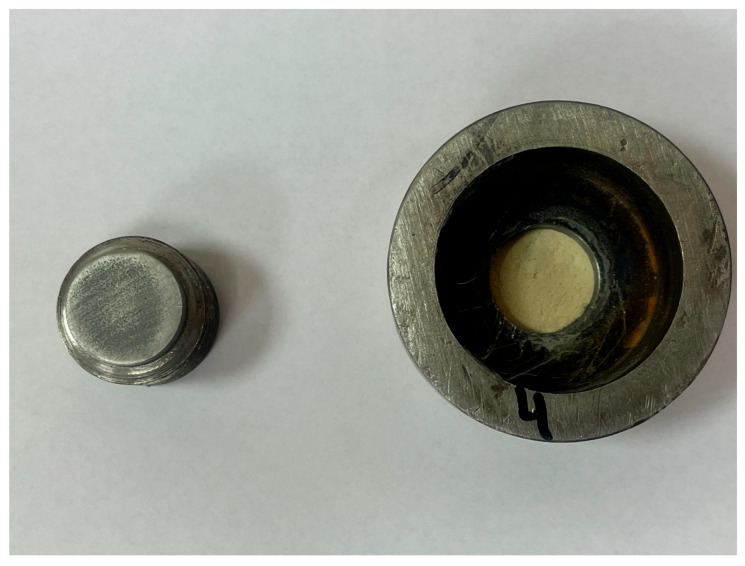
View of the sample after the pin detachment during the pull-off test.

**Table 1 materials-18-04559-t001:** Roughness profile test results.

Glass Beading																Average	Standard Deviation
R_a_	2.15	2.88	2.52	2.14	1.79	2.73	2.93	2.66	3.25	2.36	1.95	1.98	1.88	2.36	2.6	2.41	0.42
R_z_	10.8	13.5	14.4	10.8	10.4	14.1	15.6	13.6	15.3	12.1	9.91	10.2	9.43	12.1	13	12.35	1.96
P400 Sandpaper																	
R_a_	1.09	1.03	0.62	0.72	0.93	1.13	0.58	0.75	0.49	0.31	0.79	0.41	0.39	0.65	0.88	0.72	0.25
R_z_	5.7	6.03	3.99	4.28	4.44	6.44	3.2	4.47	3.04	1.94	5.4	2.57	3.18	4.47	5.42	4.30	1.29
Laser Beam Treatment																	
R_a_	1.31	1.19	1.28	1.03	0.82	1.78	1.45	1.11	1.56	1.16	1.21	1.35	1.08	1.53	1.32	1.28	0.23
R_z_	7.78	6.90	8.37	5.96	5.41	11.60	8.92	7.23	8.29	6.71	7.11	7.92	6.25	9.23	7.75	7.70	1.47

**Table 2 materials-18-04559-t002:** Average ultrasonic wave amplitude for three different surface preparation and two experimental stages.

	Laser Beam Treatment					
	Stage I			Stage II		
	Pulse _1′_	Pulse _2′_	Pulse _3′_	Pulse _1″_	Pulse _2″_	Pulse _3″_
Average Amplitude	56	32	23	42	21	13
Standard Deviation	4	4	4	3	3	2
	P400 Sandpaper					
Average Amplitude	30	18	11	23	13	7
Standard Deviation	4	2	2	3	2	1
	Glass Beading					
Average Amplitude	45	28	17	32	17	9
Standard Deviation	9	6	4	6	3	2

**Table 3 materials-18-04559-t003:** Average values of reflection coefficient.

	Laser Beam Treatment		P400 Sandpaper		Glass Beading	
	Method I	Method II	Method I	Method II	Method I	Method II
Average Amplitude	0.75	0.52	0.79	0.55	0.73	0.52
Standard Deviation	0.04	0.03	0.09	0.03	0.13	0.04

## Data Availability

The original contributions presented in this study are included in the article. Further inquiries can be directed to the corresponding author.

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
