# Peer review of "The Reflection Coefficient |r| as a Nondestructive Measure of the Coating Adhesion to a Steel Substrate"

_materials, 2025, doi:10.3390/ma18194559_

Round 1
Reviewer 1 Report
Comments and Suggestions for Authors
This manuscript has great innovative significance in investigating a non-destructive measure of coating adhesion. The work can arouse wide interests of researchers in understanding reflection coefficient |r| as a non-destructive measure of adhesion between coating and steel substrate. The manuscript is interesting. In my frank opinion, the manuscript should be deserved for its final publication in such high-level Journal. However, the main comments to be solved are as follows:
- In Introduction, noveltyand significance of this work should be further clarified. There are many non-destructive evaluations of coating adhesion to the substrate. What is distinctiveness of reflection coefficient |r|?
- In Section 2, what is design basis of the parameters for “The thickness of the coating was about 1 mm and was the same for all samples. The curing time of the coating was 25-30 min at 20°C, and during the drying of the putty layer, a gradual change in its color from greenish when applied to a sandy color after curing could be observed.”?
- If possible, please add figures for roughness profile test.
- Results of correlation test analysis are not clear. If possible, please add more discussions.
- The related reference should be added. As seen in introduction about “some solutions also provide resistance to UV radiation and are hydrophobic (provide drainage of water - no streaks and better visual effect of the paint coating) [13].”such as: Development of a mechanically robust superhydrophobic anti-corrosion coating using micro-hBN/nano-Al2O3 with multifunctional properties, Ceram. Int. 51 (2025) 491-505.” This article points out preparation of superhydrophobic coating with multifunctional properties for practical applications, hence the above reference should be quoted and added for great correlation. This article points out micro-hBN/nano-Al2O3 based superhydrophobic coating can be applied in fields of anti-fouling, self-cleaning, anti-icing, and corrosion protection for paint coating, hence the above reference should be quoted and added for great correlation.
Author Response
Dear Reviewer,
A detailed response can be found in the attached file.
Yours sincerely,
Dariusz Ulbrich

Reviewer 2 Report
Comments and Suggestions for Authors
Hello!
The authors state that comparing ultrasonic and mechanical testing is an "important innovation." However, the idea of correlating reflection coefficient |r| with adhesion strength is not novel and has been explored for a long time. Instead, the novel aspect of the study lies in comparing two specific methods for calculating |r| for a particular type of coating (NOVOL 200 HYBRYD putty) and three methods of surface preparation relevant for body repair. This should be made clearer.
It is mentioned that the thickness of the putty was "approximately 1 mm." For ultrasonic measurements, where it is crucial that reflections occur from the interface and not from the bottom of the coating, thickness is a critical parameter. However, it is not explained how the same and precise thickness was ensured and monitored across all samples. This could be a source of error.
The conclusions reflect the main findings of the study, but they are rather general. To strengthen them, specific quantitative recommendations can be added. For instance: "To assess the adhesion of a filler coating to a steel substrate non-destructively, it is recommended to use method I (full reflection) as it demonstrates a strong correlation (R² > 0.8) with mechanical strength. The optimal range for the reflection coefficient |r| of a high-quality compound after glass blasting is 0.61-0.83."
Author Response

(The authors gave the same response as above.)

Round 2
Reviewer 1 Report
Comments and Suggestions for Authors
The manuscript has been carefully revised and can be accepted.
Reviewer 2 Report
Comments and Suggestions for Authors
Thanks!